# Graph Neural Networks for Connectivity Inference in Spatially Patterned Neural Responses

**Taehoon Park**                                        GKSDIDG@HANYANG.AC.KR

**Juhyeon Kim**                                         WNGUS1310@HANYANG.AC.KR

**Donghee Kang**                                        EHDGML1312@HANYANG.AC.KR

**Kijung Yoon**                                         KIYOON@HANYANG.AC.KR

*Hanyang University, Seoul, 04763, Korea*

**Editors:** Sophia Sanborn, Christian Shewmake, Simone Azeglio, Arianna Di Bernardo, Nina Miolane

## Abstract

A continuous attractor network is one of the most common theoretical framework for studying a wide range of neural computations in the brain. Many previous approaches have attempted to identify continuous attractor systems by investigating the state-space structure of population neural activity. However, establishing the patterns of connectivity for relating the structure of attractor networks to their function is still an open problem. In this work, we propose the use of graph neural networks combined with the structure learning for inferring the recurrent connectivity of a ring attractor network and demonstrate that the developed model greatly improves the quality of circuit inference as well as the prediction of neural responses compared to baseline inference algorithms.

**Keywords:** continuous attractor networks, connectivity inference, graph neural networks

## 1. Introduction

Continuous attractor networks (CAN) have become some of the most influential circuit models in systems neuroscience. A CAN defined by a symmetric recurrent connectivity leads to spontaneous and stable neural activity patterns and the stable states can be used to represent numerous external variables such as the orientation of visual stimuli (Ben-Yishai et al., 1995), head direction (Zhang, 1996), eye position (Seung, 1996), and spatial location (Fuhs and Touretzky, 2006; Burak and Fiete, 2009). The CAN's wide range of utility has spurred several computational studies to find evidence of attractor dynamics in the brain.

The classic approach for identification of continuous attractors is to probe for the invariance of correlation structures in neural population activity (Taube et al., 1990; Yoganarasimha et al., 2006; Fyhn et al., 2007; Yoon et al., 2013; Gardner et al., 2019; Trettel et al., 2019). The invariance is an essential condition because the states of the system are internally generated by strong recurrent connections, which constrain the activity of neurons to form a certain coactivation pattern; thus, pairwise cell-cell relationships should remain invariant across time, environmental conditions, and behavioral states, independent of external sensory inputs (Burak and Fiete, 2009). Another recent attempt to address this challenge has been made to directly characterize the full state-space of the population responses and visualize them by topological data analysis (Low et al., 2018; Chaudhuri et al., 2019; Rybakken et al., 2019; Gardner et al., 2022). The resulting topological structure of

the attractor manifold provides the hints of an underlying circuit mechanism, and its invariance across the aforementioned external factors validates the main predictions of CAN models.

Nevertheless, few approaches are more reliable for identifying CAN than directly measuring the wiring diagrams of neural circuits. Hence, estimating the network connectivity from large population recordings has also been a long-standing problem (Paninski et al., 2003; Schneidman et al., 2006; Pillow et al., 2008). A fundamental challenge in circuit inference is that the mapping from circuits to activity is not injective: many neural circuits can generate the same activity states (Das and Fiete, 2020; Powell et al., 2021; Curto and Morrison, 2019). Moreover, several previous statistical models with strong parametric assumptions limit their ability to recover direct synaptic connections; thus, the inferred connectivity could be substantially different from the true connections, especially when neurons are highly correlated (Das and Fiete, 2020).

Here, we take a new approach to inferring structural connectivity with no stringent parametric assumptions based on neural spike data generated from a strongly recurrent ring attractor network. We adopt the framework of graph neural networks (GNNs) (Hamilton et al., 2017; Bronstein et al., 2017; Battaglia et al., 2018) combined with the structure learning module, an architectural design strategy that relies on learning the underlying circuitry jointly with predicting multi-neuronal spike responses. We show that this approach can be significantly more expressive than statistically sophisticated inference algorithms. Furthermore, we propose the use of an iterative neural message-passing scheme (Gilmer et al., 2017) that is closely matched to the generative nature of CAN. Taken together, these ideas suggest new possibilities for less biased inference of recurrent neural circuits.

## 2. Methods

### 2.1. Generative Model

We consider a ring network of 100 neurons where the process of generating neural spikes is by either a simple threshold-crossing or simulating an inhomogeneous Poisson process (Appendix A). The generative network model is designed to have rotation-invariant recurrent connections with a local Mexican hat weight profile and exhibit spatially periodic activity patterns drifting over time (Figure 1a)[1]. We record spikes from the network for 8 minutes with a time step $\Delta t = 0.1$ ms and represent the spike trains via $\mathbf{x} \in \{0, 1\}^{N \times L}$, where $N$ is the number of neurons and $L$ is the length of time steps (Figure 1a). The data is split into 80%, 10%, and 10% over time for training, validation and testing.

### 2.2. Inference Model

**Structure learning module**  The goal of this module is to model the pairwise connection strength $\mathbf{w}_{ij}$ given every pair of $(\mathbf{x}_i, \mathbf{x}_j)$ where $\mathbf{x}_i = (\mathbf{x}_i^1, \ldots, \mathbf{x}_i^L)$ denotes the spike train of $i$th neuron. We apply a 1D convolution $f_{\text{Conv1D}}$ with 10 kernels of size 200 [2] and a stride of 20 over each input spike train, vectorize the feature maps along time dimension, and use

---

1. This generative model is equivalent to the highly structured ring network in the strongest weight regime by Das and Fiete (2020) when the scalar weight strength parameter $r = 0.025$.

2. The kernel size is set twice as long as the synaptic time constant ($\tau = 10$ ms) of the generative model.

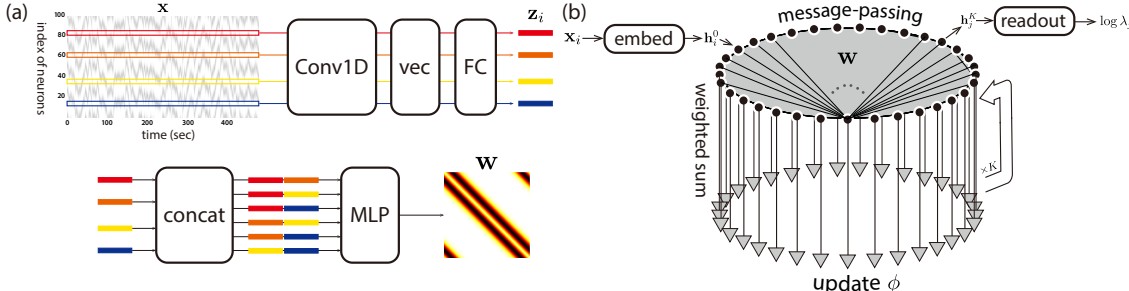

Figure 1: **Schematics of the proposed inference model. (a)** Structure learning module, **(b)** Spike prediction module.

a fully connected layer $f_{\text{out}}$ to get a reduced output embedding vector $\mathbf{z}_i$:

$$\mathbf{z}_i = f_{\text{out}}\left(\text{vec}\left(f_{\text{Conv1D}}(\mathbf{x}_i)\right)\right) \tag{1}$$

Aside from these layers, we use batch normalization (Ioffe and Szegedy, 2015) placed after `ReLU` activation function. We then concatenate every pair of $(\mathbf{z}_i, \mathbf{z}_j)$ and apply a multilayer perceptron (MLP) that has two hidden layers with 32 units to infer the strength of network connectivity:

$$\mathbf{w}_{ij} = \text{MLP}(\mathbf{z}_i \parallel \mathbf{z}_j) \tag{2}$$

The resulting inferred weight matrix $\mathbf{w} \in \mathbb{R}^{N \times N}$ is assumed to be symmetric and have no self-loop. This constraint is weaker than the rotation-invariance of target recurrent weight strengths. In simple terms, every neuron in the proposed inference model does not have to share the same outgoing synaptic weights to all of the other neurons around the ring network.

**Spike prediction module** The task of this module is to predict the spike activity $\mathbf{x}$ for every neuron given the latent circuitry $\mathbf{w}$:

$$p_\theta(\mathbf{x}|\mathbf{w}) = \prod_{t=1}^{T} p_\theta\left(\mathbf{x}^{t+1}|\mathbf{x}^t, \ldots, \mathbf{x}^1, \mathbf{w}\right) \tag{3}$$

where $\mathbf{x}^t = (\mathbf{x}_1^t, \ldots, \mathbf{x}_N^t)$ represents the spike activity of all $N$ neurons at time $t$. We assume that the attractor dynamics can be implicitly learned by modeling instantaneous neural spike rates $\boldsymbol{\lambda}^{t+1}$ where $\text{Poisson}(\boldsymbol{\lambda}^{t+1}) = p_\theta\left(\mathbf{x}^{t+1}|\mathbf{x}^t, \ldots, \mathbf{x}^1, \mathbf{w}\right)$ through the message-passing operations:

$$\mathbf{h}_{i,t}^0 = g_{\text{emb}}\left(\mathbf{x}_i^t, \ldots, \mathbf{x}_i^{t-2\tau/\Delta t + 1}\right) \tag{4}$$

$$\mathbf{h}_{i,t}^{k+1} = \phi\left(\mathbf{h}_{i,t}^k, \; \mathbf{h}_{i,t}^0 \;\middle\|\; \sum_{j \neq i} \mathbf{w}_{ij} \cdot \psi\left(\mathbf{h}_{i,t}^k \parallel \mathbf{h}_{j,t}^k\right)\right) \tag{5}$$

$$\log\left(\lambda_i^{t+1}\right) = g_{\text{out}}\left(\mathbf{h}_{i,t}^K\right) \tag{6}$$

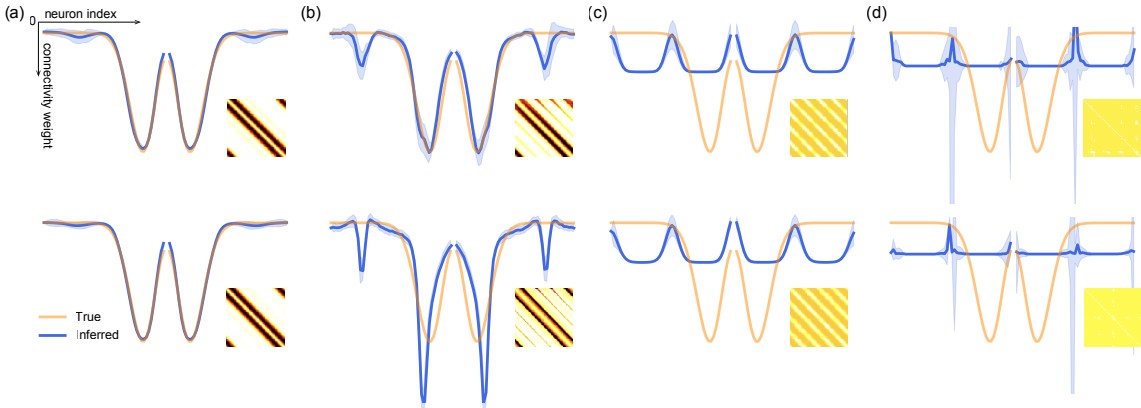

Figure 2: **Quality of circuit inference using** $4.8 \times 10^6$ **spikes from a strongly recurrent ring attractor network. (a)** True Mexican hat-shaped connectivity weight (orange) superimposed by the average inferred weight profile by GNN (blue). The inset at the bottom right presents the inferred weight matrix $\hat{\mathbf{W}}$. The model performs inference on spike data generated by threshold-crossing (top) and linear-nonlinear-poisson (bottom) models. The followings are the quality of inference by **(b)** GLM, **(c)** seqNMF, and **(d)** TCA.

where $\mathbf{h}_{i,t}^0$ is the initial embedding of the $i$th neuron's spike history for the past $\frac{\tau}{\Delta t}(= 200)$ steps at time $t$. At $k$th step of message passing, we compute the message sent from neuron $j$ to $i$ through $\psi$ and a function $\phi$ combines the sum of incoming messages weighted by connection strength $\mathbf{w}_{ij}$ with the previous latent state $\mathbf{h}_{i,t}^k$ to generate the updated neuronal state $\mathbf{h}_{i,t}^{k+1}$. After $K$ rounds of message passing, we use $g_{\text{out}}$ to read out the log-firing rate from the output of the final propagation step (for our spike trains, $K{=}1$ was enough[3]). This firing rate determines the rate of an inhomogeneous Poisson process that generates spikes in neuron $i$ at time $t + 1$. In this experiment, we use separate two-layer MLPs for $\psi$ and $g_{\text{out}}$, a fully connected layer for $g_{\text{emb}}$, and the gated recurrent unit is applied to $\phi$.

### 2.3. Training

The training loss is the negative log-likelihood of the recorded population activity and the objective is to predict the spike times accurately by minimizing the likelihood using the Adam optimizer with a learning rate of $5 \times 10^{-4}$, decayed by a factor of 0.5 every 50 epochs:

$$\mathbf{\Theta}^*, \mathbf{w} = \arg\min_{\mathbf{\Theta}} \sum_{i=1}^{N} \sum_{t=1}^{L} \left( \lambda_i^t - \mathbf{x}_i^t \log \lambda_i^t \right) \tag{7}$$

where the parameters $\mathbf{\Theta}$ are the weights of the structure learning and spike prediction network modules, and $\mathbf{w}$ is the best inferred connectivity weight matrix.

---

3. Our task is not multi-step but single-step spike predictions because the spatially patterned neural responses are not driven by an external input but drift around the ring over time. One possible issue with the single step prediction is that the inferred weights cannot have a considerable effect on short-term dynamics. Meanwhile, a large number of message-passing iterations may reduce the effect of structure learning module and for this reason we set $K$ to 1.

Table 1: Quality of spike prediction $\mathcal{L}$ and inference error $\Delta$

| generative model($\rightarrow$) | threshold-crossing | | linear-nonlinear-poisson | |
|:---:|:---:|:---:|:---:|:---:|
| inference model($\downarrow$) | $\mathcal{L}$ | $\Delta$ | $\mathcal{L}$ | $\Delta$ |
| TCA | - | 0.7619 | - | 0.7612 |
| seqNMF | - | 0.7885 | - | 0.7963 |
| GLM | 0.0409 | 0.2441 | 0.0430 | 0.4841 |
| Ours | **0.0322** | **0.0614** | **0.0334** | **0.0487** |

## 3. Experiments

To evaluate the quality of inferred connectivity, the outgoing weight profiles $\mathbf{w}$ are realigned by minimizing the gap ($\ell_1$-norm) between the estimated weights of all neurons. We then compute the average weight profile $\bar{\mathbf{w}}$ and rescale $\mathbf{w}$ by minimizing $\ell_1$ deviation between $\bar{\mathbf{w}}$ and $\mathbf{W}$ to match the scale of the true weights $\mathbf{W}$. It is necessary because the message function $\psi$ in Equation (5) may compensate for an arbitrary scale of $\mathbf{w}$. Notably, the estimated bias during rescaling is an order of magnitude smaller than the rescaling factor, which suggests that the type of inferred connectivity (i.e. excitatory or inhibitory interaction) is in good agreement with the ground truth.

We next use the following two metrics to compare the proposed approach to three baselines[4]: i) the normalized inference error, computed as $\Delta = \|\mathbf{W} - \hat{\mathbf{W}}\|_F / \|\mathbf{W}\|_F$ where $\hat{\mathbf{W}}$ is the scaled version of $\mathbf{w}$ and $\|\cdot\|_F$ denotes the Frobenius-norm, and ii) the log-likelihood score $\mathcal{L}$ in Equation (7). Every alternative inference method exhibits an error biased toward inferring unconnected neurons, evidenced by side bumps in the weight profile or by multiple off-diagonal stripes in the weight matrix (Figures 2b-2d). This type of systematic inference error is interpreted as overestimated connections resulting from highly correlated activity by strongly recurrent networks (Das and Fiete, 2020). Despite the failure to explain away strong correlations, the proposed inference model alleviates the side bumps (Figure 2a) and thus significantly reduces biased errors (Table. 1). More accurate estimation of spike activity in our model (Table. 1) suggests that the GNN-based spike prediction module is expressive enough to learn and match actual neural dynamics of a ring attractor network.

## 4. Conclusion

We presented a GNN-based neural inference model to understand the network mechanism of neural circuits by reconstruction of structural connectivity from population neural spike data. We demonstrated that the proposed model is highly effective at inferring direct connections of a strongly recurrent network. It will be interesting to test this model framework by examining real population recording data, and to see whether the inferred circuitry supports the continuous attractor dynamics in the brain.

---

4. The baselines are generalized linear model (GLM) (Pillow et al., 2008), sequence non-negative matrix factorization (seqNMF) (Mackevicius et al., 2019), and tensor component analysis (TCA) (Williams et al., 2018) (see more details in Appendix B). TCA and seqNMF are not the models for circuit inference but aim to extract low-dimensional representations and dynamics of neural data. Nonetheless, we have run those models to identify low-dimensional neuron factors (i.e. features) and obtain their correlation structure since the correlation structure offer a crude estimate of network connectivity.

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

## Appendix A. Generative network models

We construct a ring network of $N = 100$ neurons where the outgoing synaptic weights $\mathbf{W}_{ij}$ around the ring are defined by a local Mexican hat profile:

$$\mathbf{W}_{ij} = e^{-d_{ij}^2/2\sigma_1^2} - ae^{-d_{ij}^2/2\sigma_2^2} \tag{8}$$

where $d_{ij}$ is the distance between neurons $i$ and $j$, $\sigma_1 = 6.98$, and $\sigma_2 = 7$ (in units of neuron index). All inhibitory recurrent weights by setting $a = 1.0005 > 1$ allow the pattern formation and dynamical stability.

**Threshold-crossing model** The input to each neuron at time step $t$ is given by

$$\mathbf{g}(t) = r\mathbf{W}\mathbf{s}(t) + \mathbf{b}(t) \tag{9}$$

where $\mathbf{s}(t)$ is the vector of synaptic activations, $\mathbf{W}$ is the $N \times N$ matrix of recurrent connectivity defined by Equation (8), and $\mathbf{b}(t) = 10^{-3}(1 + \xi(t))$ is the feed-forward inputs. $\xi(t)$ is a white Gaussian noise per neuron, with zero mean and s.d. $\sigma_\xi = 0.3$, resulting in a Poisson-like variance proportional to the mean activation. We set the weight strength $r = 0.025$. The neurons in this generative model emit a spike when the input $g_i(t)$ to neuron $i$ at time step $t$ exceeds a threshold $\theta$. The synaptic activation from neurons that just spiked is incremented by 1 and otherwise decays exponentially with time constant $\tau = 10$ ms according to the following equation:

$$\mathbf{s}(t + \Delta t) = \mathbf{s}(t)\left(1 - \frac{\Delta t}{\tau}\right) + \sigma(t) \tag{10}$$

where $\sigma(t)$ is the binary vector of spikes from the network.

**Linear-nonlinear Poisson model** The inputs to this model are determined by Equation (9) except that the external input $\mathbf{b}(t) = 10^{-3}$ that is constant with no noise. The neural firing rate $\lambda_i(t)$ is determined by passing the summed input $g_i(t)$ to the rectifying nonlinear activation function:

$$\lambda_i(t) = \lambda_0 \, \mathtt{ReLU}\left[g_i(t) - \theta\right] \tag{11}$$

These firing rates determine the rate of an inhomogeneous Poisson process that generates $n_i(t)$ spikes in neuron $i$ at time $t$.

$$n_i(t) \sim \mathbf{Pois}(\lambda_i(t)) \tag{12}$$

The synaptic activations are determined by Equation (10) except that $\sigma_i(t)$ is replaced by $n_i(t)$.

## Appendix B. Baseline inference algorithms

---

**Algorithm 1** Generalized Linear Model

---

1  Initialize the model parameters by setting $b = 0, \mathbf{w} = 0, \mathbf{z} = 1$

2  **for** $i \in [1, N]$ **do**

3      **for** $k \in [1, 2000]$ **do**

4          Coupling filter $f_{\mathrm{couple}} = \mathbf{Bz}$

5          **if** $i > 1$ **then**

6              circular shift $\mathbf{w}$ by $i$ steps

7          **end if**

8          $\boldsymbol{X}_{\mathrm{neigh}} = \mathbf{x}_{j \in N(i)}^{1:L-1}$

9          $\log \lambda_i = f_{\mathrm{couple}} * (\boldsymbol{X}_{\mathrm{neigh}} \mathbf{w}) + b$

10          $\mathcal{L} = \sum_t \sum_i \left( \lambda_i^t - \mathbf{x}_i^t \log \lambda_i^t \right)$

11          $b, \mathbf{w}, \mathbf{s} = \underset{b, \mathbf{w}, \mathbf{s}}{\arg \min}(\mathcal{L})$ by Quasi-Newton Method

12      **end for**

13  **end for**

---

We first arrange a $200 \times 10$ matrix $\mathbf{B}$ whose column represents a raised cosine filter. The coupling filter $f_{\mathrm{couple}}$ in GLM is constructed by a linear combination of the basis $\mathbf{B}$. Then the log-firing rate $\lambda_i$ is computed by linearly projecting neighbored neural spike trains onto $f_{\mathrm{couple}}$ weighted by the connection strengths $\mathbf{w}$. All the model parameters are updated by Quasi-Newton method.

---

**Algorithm 2** Tensor Component Analysis (TCA)

---

1 Initialize $\mathbf{N}, \mathbf{K}, \mathbf{T}$ (neuron, trial, and time factors respectively)

   $\mathbf{N} \in \mathbb{R}^{N \times R}, \mathbf{K} \in \mathbb{R}^{K \times R}, \mathbf{T} \in \mathbb{R}^{T \times R}$

   The number of neurons $N$, components $R$, total time steps $T$, length of sequence $L$

2 $\mathbf{M} = \mathrm{EW}\,(\mathbf{X})$

   EW: exponentially weighted moving average

3 Reshape $\mathbf{M} \in \mathbb{R}^{N \times L}$ to $\mathbb{R}^{N \times K \times T}$, $T = L/K$

4 **for** $k \in [1, 500]$ **do**

5      $\mathbf{N}_{:,r}, \mathbf{K}_{:,r}, \mathbf{T}_{:,r} = \underset{\mathbf{N}_{:,r}, \mathbf{K}_{:,r}, \mathbf{T}_{:,r}}{\arg\min}\ (\|\mathbf{M} - \sum_r \mathbf{N}_{:,r} \otimes \mathbf{K}_{:,r} \otimes \mathbf{T}_{:,r}\|_F)$,

   by alternating least-squares method

6 **end for**

7 **for** $r \in [1, R]$ **do**

8      $\mathbf{Z}^{(r)} = \mathbf{N}_{:,r} \otimes \mathbf{N}_{:,r} \otimes \mathbf{K}_{:,r} \otimes \mathbf{T}_{:,r}$

9      $\mathbf{E}^{(r)} = \sum_{t=1}^{T} \sum_{k=1}^{K} \left( \mathbf{Z}^{(r)}_{:,:,k,t} \right)$

10 **end for**

11 $\mathbf{E} = \frac{1}{R} \sum_r \left( \mathbf{E}^{(r)} \right)$

---

TCA decomposes an input tensor into $R$ components of neuron, trial, and time factors respectively. We convolve spike trains with an exponential filter, then the filtered spike trains are reshaped to have $K = 10$ trial components. Next, we apply TCA and take the outer product of each neuron factor $\mathbf{N}_{:,r}$ with itself followed by multiplying trial and time factors for estimating low-dimensional average correlation structure over trial and time.

---

**Algorithm 3** Sequence Non-negative Matrix Factorization (seqNMF)

---

1  Initialize neuron factor $\mathbf{W}$ and time factor $\mathbf{H}$

   $\mathbf{W} \in \mathbb{R}^{N \times R \times L}$, $\mathbf{H} \in \mathbb{R}^{R \times T}$

   The number of neurons $N$, components $R$, total time steps $T$, length of sequence $L$

2  Define smoothing matrix $\mathbf{S} \in \mathbb{R}^{T \times T}$

   $\mathbf{S}_{i,j} = 1$ when $|i - j| < L$ otherwise 0

3  Define regularization parameter $\beta$

4  **for** $k \in [1, 100]$ **do**

5      $\tilde{\mathbf{X}}_{i,t} = \sum_r \sum_l \mathbf{W}_{i,r,l} \mathbf{H}_{r,(t-l)}$

6      $\mathbf{C}_{i,j} = \sum_l \sum_n \mathbf{W}_{n,i,l} \mathbf{X}_{n,j+l}$

7      $\mathbf{W}, \mathbf{H} = \underset{\mathbf{W},\mathbf{H}}{\arg\min} \left( \left\| \tilde{\mathbf{X}} - \mathbf{X} \right\|_F + \beta \left\| \mathbf{C} \cdot \mathbf{S} \cdot \mathbf{H}^\top \right\|_{1, i \neq j} \right),$

   by non-negative gradient descent (Lee and Seung, 1999, 2000)

8  **end for**

9  **for** $r \in [1, R]$ **do**

10      $\mathbf{Z}^{(r)} = \mathbf{W}_{:,r,0} \otimes \mathbf{W}_{:,r,1} \otimes \mathbf{H}_{r,:}$

11      $\mathbf{E}^{(r)} = \sum_{t=1}^{T} \left( \mathbf{Z}^{(r)}_{:,:,t} \right)$

12  **end for**

13  $\mathbf{E} = \frac{1}{R} \sum_r \left( \mathbf{E}^{(r)} \right)$

---

SeqNMF is a dimensional reduction method with regularized convolutional non-negative matrix factorization to extract sequential patterns. We first define the smoothing matrix $\mathbf{S} \in \mathbb{R}^{T \times T}$ with the length of sequence $L = 2$ and regularization parameter $\beta = 0.001$, and then compute the neuron factor $\mathbf{W} \in \mathbb{R}^{N \times R \times L}$ and time factor $\mathbf{H} \in \mathbb{R}^{R \times T}$ by Algorithm 3, We take the outer product between learned neuron factors multiplied by time factor $\mathbf{H}_{r,:}$ to extract the average component-wise edge features as in TCA.

