# OpenReview forum: "Graph Neural Networks for Connectivity Inference in Spatially Patterned Neural Responses"
_NeurIPS.cc/2022/Workshop/NeurReps — NeurReps 2022 Poster_

### Official Review · Reviewer_MF9n · 2022-10-10
**Interesting technical contribution, but generality of the model are unclear due to insufficient experimentation**

**Confidence:** 4
**Soundness:** 3
**Presentation:** 4
**Contribution:** 2
**Overall Rating:** 5

**Summary:**

The paper introduces a new model that can more effectively predict spiking activity from neurons in a continuous attractor network. It does so by inferring the structure of the network underlying the data in a preliminary module, then using that structure in a message-passing graph neural network to approximate firing rates over time. They demonstrate that this system can accurately infer the wiring diagram of a classic ring network with the Mexican hat weight profile, much better than previous methods.

**Questions:**

In the experiments section, the authors describe realigning and rescaling the inferred weight profiles to match the true weights. To me, this seems like the solution of the circuit needs to be known in advance to examine the model's accuracy. If my interpretation is true, this requirement damages the model's generality significantly. It would be useful to clarify this, and if I am correct, the approach should be rethought.

In the spike prediction module, the authors write that they only used 1 round of message passing in the GNN. Is this not simply due to the one-step input-output dependence in the neural network under the simple Poisson process model neuron? That is, only the direct connections between neurons matters for describing spiking. This may not be true in the case of model neurons with intrinsic "memory" like LIF neurons. That leads me to question the implementation of your neuron models. I would appreciate seeing the technical descriptions of the model neurons in the appendix, because if your "threshold-crossing" model is in fact an LIF, the fitting of K=1 would be interesting and unexpected.

**Limitations:**

As mentioned in the weaknesses, the model seems useful for inferring "hard-coded" circuits like those in drosophila or c. elegant, but remain in question for distributed CANs like those that may exist in cortex. I would like to see how this model generalizes to circuits like those.

Given the complexity of the model design, this paper is limited by the length of the extended abstract track, and probably would have been better off in the proceedings track with more experiments.

This model also seems to be limited by assuming that a neural network is isolated from any input, which is not true in biological CANs. I am curious to see how they would generalize their model to take account for this fact.

**Recommended Decision:**

2: Borderline

**Relevance:**

3: Solid fit

**Strengths And Weaknesses:**

The authors' technical description of the model is very detailed and helped me gain insight on its function. The separation of the model into the structure inference and spike prediction module is innovative and strongly motivated by the biology. The use of the GNN is strongly motivated. In general, the model design is strong. I believe that the paper is original, well technically described, organized, and motivated. The paper could be of importance, but I am uncertain because there aren't enough validation experiments and the problem may be ill-defined.

I urge the authors to review and consider citing both Eve Marder and Carina Curto's work on the relationship between structure and function. One of the authors' claims is that they can infer circuit diagrams from spike trains, but Marder has succinctly showed that many neural architectures can implement the same dynamics. The authors' implementation of the CAN attractor is the simplest possible. Thus it remains in question as to whether the model can learn the circuitry of a CAN not in a privileged, sparse basis of neurons.

As far as testing CANs, there are many other network implementations that are simple that could be tested. One example that came to mind is a simple linear dynamical system with one eigenvalue at 0 (thus, a line attractor). The model should be easily able to infer the circuit diagram of a system like this too despite the line attractor being distributed across neurons at all times. Testing the model's capacity for line attractors in many random directions would be an easy test for generality that the authors could add to strengthen the paper.

Additionally, I am not certain that the seqNMF and TCA models are useful comparisons (and perhaps the GLM) for circuit inference. As far as I'm aware, they do not claim to be able to do circuit inference, but rather latent dynamics inference which is not strictly dependent on circuitry (as mentioned above). However, I did appreciate their consideration of multiple different models.

In general, I don't think there are enough experimental results in the paper,  especially in considering different CAN implementations. Another figure with another experiment would be highly desired.

**Submission Track:**

Extended Abstract (4 Page)

---

> ### Author Response · Authors · 2022-11-02
> **Response to Reviewer MF9n**
>
> We thank the reviewer for their support, and insightful thoughts. We appreciate your feedback and address your comments below:
>
> *"... Thus it remains in question as to whether the model can learn the circuitry of a CAN ..."*
>   - You are exactly right. The surjective mapping from circuits to activity is a fundamental problem of circuit inference; it is known that one can better estimate the connectivity from neural data when the inference model exactly matches the generative model, which is not true in general. This work is motivated by the necessity to reduce the mismatch between the true network dynamics and the model used for inference.
>   - Thus GNN-based inference model was presented with hoping that the message-passing scheme is able to learn true underlying dynamics purely from spike data. The superiority of our approach over GLM when it comes to predict spiking activity in ring attractor networks suggests that GNNs may play a pivotal role in inferring the network connectivity.
>   - Neverthelss, we agree that the use of many other generative models can help us test the effectiveness of our proposed framework. We are actively working on testing the model given simulated and real spike data that support 1D/2D attractor dynamics.
>   - Thank you for sharing the insightful references. We have added the following two papers for related works.
>     - https://doi.org/10.1016/j.cub.2021.08.042
>     - https://doi.org/10.1016/j.conb.2019.06.003
>
> *"... not certain that the seqNMF and TCA models are useful comparisons ..."*
>   - Yes, TCA and seqNMF are not the models for circuit inference but aim to extract low-dimensional representations and dynamics of neural data. Nonetheless, we have run those models to identify low-dimensional neuron factors (i.e. features) and obtain their correlation structure since it offers a crude estimate of network connectivity. We applied those available methods with a slight modification to perform simple correlation-based inference.
>   - We have updated Section 3 and Appendix B to clarify why we chose the baseline models as a meaningful comparison for circuit inference.
>
> *"In the experiments section, the authors describe realigning and rescaling the inferred weight ..."*
>   -  While we agree that the current metric for inference error requires a target weight profile, the realignment is simply done by minimizing the gap ($\ell_1$-norm) between the estimated weights of all neurons: target weights are not needed in this step as long as the inferred connectivity is expected to have a translation/rotation invariance property as in typical attractor networks.
>   -  Indeed we confirm that the inferred weight matrix is close to circulant and that the outgoing weight profiles of all neurons share the same scale and sign even without the rotation-invariance constraint of recurrent weight strengths. We also find that the estimated bias during rescaling is an order of magnitude smaller than the rescaling factor, which suggests that the type of inferred connectivity (i.e. excitatory or inhibitory interaction) is in good agreement with the ground truth.
>   -  Results indicate that our model allows the specification of overall network connectivity patterns, which can strongly narrow down the underlying network dynamics even without the target weights.
>
> *"In the spike prediction module, the authors write that they only used 1 round of message passing ..."*
>   - We thank the reviewer for this question. Below we provide clarifications regarding the message-passing round.
>     - Our training loss involves single-step spike predictions. We decide not to make multi-step predictions because the spatially patterned neural responses are not driven by an external input but drift around the ring over time.
>     - One possible issue with the single step prediction is that the inferred weights cannot have a considerable effect on short-term dynamics. Meanwhile, a large number of message-passing iterations may reduce the effect of structure learning module and for this reason we set $K$ to 1.
>     - We are currently working on another experiment where the system is driven by an external input to understand how multi-step predictions affect the accuracy of connectivity inference as a function of $K$.
>   - We have added the technical descriptions of the model neurons in the appendix A.
>
> *"... but remain in question for distributed CANs like those that may exist in cortex. I would like ..."*
>   - Thank you for the stimulating suggestion. We will think about it for future work.
>
> *"... limited by assuming that a neural network is isolated from any input, ..."*
>   - Thank you for pointing this out. We are currently testing on the populatin activity recorded from the attractor system driven by an input signal. Our framework is flexible enough to take and process any input for the same task, thus the curent experimental setup does not limit the model's applicability.

---

### Official Review · Reviewer_oXqb · 2022-10-15
**A strong contribution towards better inference of network connectivity in continuous attractor networks.**

**Confidence:** 4
**Soundness:** 4
**Presentation:** 2
**Contribution:** 4
**Overall Rating:** 7

**Summary:**

Continuous attractor networks form a very popular class of models in computational neuroscience. A fundamental problem of interest is estimating the network connectivity from large population recordings. Previous attempts to infer the structural connectivity have been shown to be heavily biased, producing high systemic error, especially for strongly recurrent networks. The authors propose a GNN-based method consisting of an inference model to infer the connectivity weight matrix and a spike prediction module. They introduce metrics to compare the proposed method to three different baselines.

**Questions:**

Suggestions: I would highly suggest a rewriting of the methods section, especially the section describing the spike prediction module. There are some typos, such as the expression for the Poisson distribution (depending on "z", should be "w"). This section is too compressed, and the final paragraph very difficult to get through. An illustration of the architecture and better description of the various functions used would drastically improve this section.

**Limitations:**

This paper adequately addresses the limitations of presented work.

**Recommended Decision:**

3: Accept

**Relevance:**

3: Solid fit

**Strengths And Weaknesses:**

This extended abstract offers valuable contributions to the topic. I consider this a fairly strong paper, which makes a good argument for further exploration of this GNN-based method for neural connectivity inference. A point of strength is that this method relaxes the rotation-invariance constraint of recurrent weight strengths used in previous analyses. The resulting circuit inference is shown to be significantly better than that made by other methods, namely GLM, seqNMF, and TCA, each of which infers nonexistent connections between neurons in a strongly recurrent network. The performance of the this method, as measured by the proposed metric, is quite impressive compared to the older methods.

A point of weakness for this paper is the clarity of the writing. The description of the spike prediction module is particularly messy.

**Submission Track:**

Extended Abstract (4 Page)

---

> ### Author Response · Authors · 2022-11-02
> **Response to Reviewer oXqb**
>
> Thank you for the encouraging comments, and valuable suggestions. We appreciate your feedback and address your comments below:
>
> *"The description of the spike prediction module is particularly messy."*
>   - Thank you for expressing your concern. We have reworded the methods section to ensure that it conveys the message of the paper well.
>
> *" There are some typos, such as the expression for the Poisson distribution ..."*
>   - There was a writing error, and we apologize for that. We have edited the expression for the Poisson distribution in section 2.2 to reflect the conditioning latent circuitry "w" rather that "z".
>
> *"An illustration of the architecture and better description of the various functions ..."*
>   - We have added Figure 1: the proposed model framework of GNNs combined with the structure learning module.

---

### Official Review · Reviewer_8g8a · 2022-10-17
**review of Graph Neural Networks for Connectivity Inference in Spatially Patterned Neural Responses**

**Confidence:** 3
**Soundness:** 3
**Presentation:** 3
**Contribution:** 3
**Overall Rating:** 7

**Summary:**

This contribution decribes a GNN to estimate the connectivity in attractor networks, in the goal of matching the observed functional behavior of an attractor model with the corresponding underlying network structure.


**Questions:**

It could be interesting to have a figure that summarizes the proposed methods, with the different modules. Even if the space is limited, this is perfectly fine to be in the appendix.


**Limitations:**

The experiments demonstrate qualitative results, but the generative model is limited in terms of number of neurons. As the long-term objective is to address the problem of estimating the connectivity from large population recordings, it could be interesting to see how the GNN behaves when the number of neurons scales up.


**Recommended Decision:**

3: Accept

**Relevance:**

3: Solid fit

**Strengths And Weaknesses:**

Continuous Attractor Networks (CAN) are widely used theoretical models in computational neuroscience, with the limitation that inferring the connectivity from observation is a difficult problem. The proposed GNN model aims to uncover this connectivity, with a novel neural message-passing function specific to spike trains.

The results are convincing for a small controlled example, with two different spike generation schemes. Indeed, it could be interesting if those results generalize to other generative models.


**Submission Track:**

Extended Abstract (4 Page)

---

> ### Author Response · Authors · 2022-11-02
> **Response to Reviewer 8g8a**
>
> Thank you for your thoughtful comments and suggestions. We appreciate your feedback and address your comments below:
>
> *"Indeed, it could be interesting if those results generalize to other generative models."*
>    - We agree that testing on various generative models can help us understand the generalization capability of the proposed model. This is something we are actively thinking about for future work in terms of simulated or real physiological data that support 1D/2D attractor dynamics.
>
> *"It could be interesting to have a figure that summarizes the proposed methods ..."*
>    - We have added Figure 1: the proposed model framework of GNNs combined with the structure learning module.
>
> *"... but the generative model is limited in terms of number of neurons. ... it could be interesting to see how the GNN behaves when the number of neurons scales up."*
>    - Thank you for pointing this out. We agree that extending the study toward larger networks -- such as ring attractors with several hundreds of neurons or two-dimensional CANs with ~10^4 neurons -- is an important next step. We are currently working on extending the generative model to a larger scale.
>    - Although we estimate the weight matrix for all neuron pairs, the number of model parameters does not grow quadratically with the number of neurons. The computations in structure learning module can be parallelized as well.

---

### Decision · Program_Chairs · 2022-10-21

Accept (Poster)